# Investigating the Potential of Network Optimization for a Constrained Object Detection Problem

**DOI:** 10.3390/jimaging7040064

**Published:** 2021-04-01

**Authors:** Tanguy Ophoff, Cédric Gullentops, Kristof Van Beeck, Toon Goedemé

**Affiliations:** EAVISE, PSI, KU Leuven, Jan Pieter De Nayerlaan 5, 2860 Sint-Katelijne-Waver, Belgium; cedricgullentops@hotmail.com (C.G.); kristof.vanbeeck@kuleuven.be (K.V.B.); toon.goedeme@kuleuven.be (T.G.)

**Keywords:** object detection, single-shot, embedded devices, mobile convolutions, depth-wise separable convolutions, pruning, quantization

## Abstract

Object detection models are usually trained and evaluated on highly complicated, challenging academic datasets, which results in deep networks requiring lots of computations. However, a lot of operational use-cases consist of more constrained situations: they have a limited number of classes to be detected, less intra-class variance, less lighting and background variance, constrained or even fixed camera viewpoints, etc. In these cases, we hypothesize that smaller networks could be used without deteriorating the accuracy. However, there are multiple reasons why this does not happen in practice. Firstly, overparameterized networks tend to learn better, and secondly, transfer learning is usually used to reduce the necessary amount of training data. In this paper, we investigate how much we can reduce the computational complexity of a standard object detection network in such constrained object detection problems. As a case study, we focus on a well-known single-shot object detector, YoloV2, and combine three different techniques to reduce the computational complexity of the model without reducing its accuracy on our target dataset. To investigate the influence of the problem complexity, we compare two datasets: a prototypical academic (Pascal VOC) and a real-life operational (LWIR person detection) dataset. The three optimization steps we exploited are: swapping all the convolutions for depth-wise separable convolutions, perform pruning and use weight quantization. The results of our case study indeed substantiate our hypothesis that the more constrained a problem is, the more the network can be optimized. On the constrained operational dataset, combining these optimization techniques allowed us to reduce the computational complexity with a factor of 349, as compared to only a factor 9.8 on the academic dataset. When running a benchmark on an Nvidia Jetson AGX Xavier, our fastest model runs more than 15 times faster than the original YoloV2 model, whilst increasing the accuracy by 5% Average Precision (AP).

## 1. Introduction

Deep learning has proven to be successful in a wide variety of applications, with many computer vision tasks such as image classification [1], image segmentation [2], object detection [3,4,5], etc., all adopting neural networks as the de facto standard for solving them. In order to leverage these techniques in commercial systems, it is necessary to be able to deploy deep neural networks on many different systems with limited computational resources, whilst still ensuring (near) real-time performance. Deep learning is, however, quite notorious for requiring lots of computational power. Especially with the general trend of developing deeper and more complex networks to increase the accuracy on well-established academic datasets [6,7,8]. These datasets are often challenging, in order to compare many state-of-the-art techniques with each other and establish the most successful ones. They include very diverse images in terms of composition, distance, viewpoint, object variety, etc. This is in stark contrast with many real-life use-cases, which are almost always somewhat constrained. They might contain specific scene constraints, such as a fixed viewpoint or similar color scheme, or the problem on its own can be simple, e.g., a one-class detection only. This very fact actually simplifies the task for the neural network significantly, but in practice it is often not feasible to acquire enough data, in order to design and train smaller, task-specific networks. A common methodology to cope with this lack of data is to use transfer learning [4,9]. Here, we first train a model on a larger dataset, and then use those weights as a starting point to train the same model on a different, usually smaller, dataset. A downside of transfer learning is that the same model is used for both datasets and thus the network might be over-engineered for the operational dataset.

Recently, the optimization of existing networks has been extensively researched. Such techniques aim to reduce the computational complexity with (almost) negligible impact on the accuracy. Examples of these techniques are depth-wise separable convolutions [10,11,12], pruning [13,14], knowledge distillation [15], layer fusion [16], weight quantization [16,17], etc. However, these techniques are often only tested in isolation and on classification datasets. This last year, there has been more research interest into object detection optimization studies for operational use cases. Wu et al. [18] looked at pruning single-shot object detectors for the operational case of apple flower detection, but they only researched the pruning optimization technique on a single dataset. Ayob et al. [19] investigated the combination of depth-wise separable convolutions and pruning for underwater object detection.

In this paper we propose a pipeline that combines depth-wise separable convolutions, pruning and weight quantization. We validate our pipeline for the task of object detection and evaluate our models on two completely different datasets, both situated at opposite extrema of the problem constrainedness spectrum. First, we performed our tests on the academic Pascal VOC dataset [7]. We investigated how much we can reduce the computational complexity of our models without deteriorating the accuracy. These results served as our baseline. Afterwards, we performed the same tests on the constrained operational Long Wave Infrared (LWIR) dataset [20]. By comparing these results we demonstrate that the complexity of existing networks can be reduced significantly.

There are two different approaches to deep learning based object detection: two-staged and single-shot detectors. Two-staged approaches such as the Region based Convolutional Neural Network detector (R-CNN) [3] first generate a number of bounding boxes around potential objects. These so-called region proposals then go through a neural network, in order to be validated as an actual object and classified into one of the categories of the dataset. Since each potential cutout is evaluated separately, these types of detection networks tend to be quite slow. Fast-RCNN [21] and Faster-RCNN [22] made significant improvements to this approach in two manners. Firstly, the computations are shared whilst classifying the different objects. Secondly, a deep learning based region proposal method is used, which reduces the number of false positives. However, these networks still remain orders of magnitude slower than single-stage approaches. Indeed, by re-framing the object detection problem as a regression of coordinates, single-shot detectors are able to simultaneously detect and classify multiple objects in an image using only a single network. As we target devices with constrained resources in this work, we decide to focus on these single-shot detectors, such as You Only Look Once (Yolo) [4,5], Single Shot MultiBox Detector (SSD) [23], Cornernet [24], etc.

In this paper we use the lightweight YoloV2 detector as our baseline [4], as it provides an excellent speed-accuracy trade-off for embedded devices. We applied different techniques to reduce the computational complexity and size of our model even further, whilst still maintaining the original accuracy. We performed these tests on an established academic dataset, Pascal VOC [7], as well as on a constrained operational dataset, the LWIR Railway Surveillance Data [20]. These two sets allow us to evaluate and compare the efficiency of our proposed techniques in more realistic scenarios. Finally, we benchmarked our models in order to validate whether we can reach the required real-time performance on an embedded device, an Nvidia Jetson AGX Xavier.

Our main contributions are:We provide a new backbone for YoloV2, by swapping out all convolutions for depth-wise separable convolutions. All of our new models (YoloV2Upsample, MobileYoloV2, MobileYoloV2Upsample; see Section 2.2) are available in our open-source library, Lightnet [25].We combine the optimization techniques of hard pruning and weight quantization to investigate the achievable amount of optimization for an object detector. Specifically, we study the influence of the constrainedness of the dataset on this optimizability. We conclude that for constrained operational object detection datasets, much larger optimization factors can be achieved than for general benchmark datasets from academic challenges.We perform an ablation study, in order to compare the influence of each individual optimization method.

## 2. Materials and Methods

In this section we discuss the chosen datasets (Section 2.1) and explain the implemented techniques we employed to reduce the computational complexity of our baseline network, YoloV2 (Section 2.2, Section 2.3 and Section 2.4).

### 2.1. Datasets

The main goal of this research is to investigate the potential of these network optimization techniques, which are developed on academic datasets, when applied to operational datasets. For this, we apply our techniques on two typical use-cases: both an academic dataset, Pascal VOC [7], and an operational dataset, the LWIR Railway Surveillance Data [20] and investigate in each of these two cases how much they can be slimmed down.

Pascal VOC is a dataset for object detection, which contains 21,503 images with 20 different classes. As seen in Figure 1a, the various different classes are completely unrelated (e.g., “aeroplane”, “cow”, “person”, “tvmonitor”), which is often seen in an academic dataset and makes detection on these data much more difficult. For our experiments, we used the Pascal VOC 2007+2012 dataset and combined their “training 2007”, “training 2012” and “validation 2012” splits for training, whilst using the “validation 2007” split for validation. Because the annotations for the VOC 2012 testing set are not released publicly, we only use the ones from 2007 for testing purposes. The number of images per split are shown in Table 1.

The publicly available LWIR Railway Surveillance Data (https://iiw.kuleuven.be/onderzoek/eavise/viper/dataset, accessed on 31 March 2021) is a dataset for person detection in long wave infrared videos. The single class nature and fixed camera viewpoint are two common scene constraints in real-life scenarios, which makes this a prime example of this kind of operational dataset (see Figure 1b). Moreover, comparing the person cutouts in Figure 2 shows that there is much less intra-class variance; i.e., the academic Pascal VOC data contain images where persons are annotated from many different viewpoints and distances, compared to the LWIR data where all persons have a similar appearance, size and viewpoint. The dataset consists of 21,852 frames split across 28 different video sequences. Since the original paper for this dataset does not provide train, validation and test splits, we need to create our own. Care must be taken when splitting video sequences to not put frames of the same sequence in different splits, as the models could then overfit to the data unnoticeably. We thus split the 28 different sequences into 3 subsets, trying to match the 65-10-25% split from VOC. The video sequences in each split are listed in Table 2 and the number of images in each split are found in Table 1.

Our operational dataset contains many scene constraints and is much less challenging than the Pascal VOC data. This is done on purpose as it allows us to cover two completely different kinds of dataset of various complexity. By validating our techniques on these distinctive datasets, we make a strong case for the generalizability of our optimization pipeline.

### 2.2. Depth-Wise Separable Convolutions

A first technique that can be used in order to reduce the number of computations, is to replace all regular convolutions in the network with depth-wise separable convolutions. Initially introduced by Sifre and Mallat [10] and later popularized by Howard et al. [11] in their MobileNet paper, depth-wise separable convolutions are a form of factorized convolution that split a regular convolution into a depth-wise and point-wise convolution (see Figure 3). A standard convolution both filters and combines information from multiple previous feature maps in a single step, resulting in the following computational cost:(1)$$D_{k}\cdot D_{f}\cdot M_{i}\cdot M_{o}$$
where $D_{k}$ and $D_{f}$ are the dimension (width × height) of the kernel and feature map, respectively, and $M_{i}$ and $M_{o}$ are the depth of the input and output feature maps.

Depth-wise separable convolutions split this into two distinct operations, the depth-wise convolution for applying a filter and a point-wise convolution for combining information of multiple filters. This results in a total computational cost:(2)$$D_{k}\cdot D_{f}\cdot M_{i}+D_{f}\cdot M_{i}\cdot M_{o}$$

Notice that both regular and depth-wise separable convolutions combine the same information and generate an output with the same shape. However, the computational cost of depth-wise separable convolutions is clearly lower:(3)$$\frac{D_{k}\cdot D_{f}\cdot M_{i}+D_{f}\cdot M_{i}\cdot M_{o}}{D_{k}\cdot D_{f}\cdot M_{i}\cdot M_{o}}=\frac{1}{M_{o}}+\frac{1}{D_{k}}<1$$

Our baseline network, YoloV2, is a fully convolutional network, where each convolution uses a kernel size of 3 × 3 ($D_{k}=9$). As the number of output feature maps in that network are always orders of magnitude bigger than nine, we can thus expect our network to have around nine times less computations, when swapping regular convolutions by depth-wise separable convolutions. However, as seen in Section 3, replacing regular convolutions with depth-wise separable convolutions results in a significant drop of Average Precision (AP). Indeed, depth-wise separable convolutions have a more restricted modeling capability compared to regular convolutions and thus a drop in accuracy might be expected.

We determined experimentally that both the first and last convolutions work best with these extra modeling capabilities. We thus decided to keep regular convolutions for the first convolution of the network, as well as the second to last convolution. The latter combines information from 2 feature maps from different parts of the network (see Figure 4). We coined this optimized architecture *MobileYoloV2*. Whilst significantly faster, this architecture still presents a notable reduction in accuracy compared to the original network.

One of the easiest ways to improve the performance of single-shot detectors is to increase the input resolution of the images going into the network. However, this also increases the amount of computations by a significant amount (see Table 3). Instead, we reduce the amount of downsampling that the model performs. This ensures that the output resolution of the model is bigger and thus the model is able to perform better. To limit the impact on the computational performance, we modify the network as close to the end as possible. We therefore choose to increase the output resolution, by removing the *“reorg”* operation introduced before concatenation of 2 feature maps of different dimensions. Instead we upsample the smallest feature map (see Figure 5). Our modification does not change the aim of the concatenation operation, which is to combine shallow, but spatially fine-grained features with higher level, more downsampled features (see Figure 4). Instead of chopping down the spatially bigger feature map, we upscale the smaller one. This results in spatially bigger feature maps with more fine-grained details, allowing for potentially better detection results and/or localization. As the concatenation happens at the end of the network, with only two convolutions remaining, the influence on the computational complexity remains rather limited (see Table 3). In fact, as both YoloV2 and MobileYoloV2 have the same last 2 layers—a regular convolution followed by a last point-wise convolution—the computational overhead of upsampling is exactly 6.11 Giga Multiply-Accumulate operations (GMAC) for both networks. We named these models with upsampling *YoloV2 Upsample* and *MobileYoloV2 Upsample*, respectively.

### 2.3. Channel-Wise Pruning

A second method to reduce the computational complexity of a neural network is to reduce the number of channels of the intermediate convolutions and feature maps. This method is called pruning and it relies on the fact that networks tend to be highly overparameterized [26,27]. However, contemporary experience seems to indicate that it is easier to train overparameterized networks [15,27,28]. Pruning exploits this fact, by removing redundant and low importance filters from a trained network. When training models on small operational datasets, we use transfer learning in order to adapt a pretrained network to our specific use-case. When retraining, we want to keep the complex modeling capabilities that the original network contains. Afterwards pruning allows us to remove the redundancy in the network, which is much more present in the case of constrained operational datasets.

Since our network is fully convolutional and we aim to run the network on a Graphics Processing Unit (GPU), we focus on the channel-wise pruning of convolutional filters. Many pruning implementations emulate pruning by masking or replacing kernel weights with zeroes [29,30,31]. While this is a completely valid approach which can help to reduce the size of the weight files, it does not reduce the number of computations when implemented on GPUs. Our framework effectively removes the channels from the convolutions during the pruning step, reducing the computational complexity of the network. As shown in Figure 6, when you remove a channel from a convolution, this modifies the dimension of the output feature map. This in turn influences the next operation that the network will perform with that feature map. This is trivial for the simple case of a linear sequence, depicted in Figure 6, but requires careful dependency tracking when working with more complex networks. Hence, we implemented a generic convolutional pruning framework in our open source Lightnet library [25] for PyTorch [29], which generates a dependency tree of the operations after each convolution and then takes care of adapting these operations. Most network architecture add a batch normalization layer and a non-linear activation after each convolution. The normalization layer contains parameters specific for the different channels of the feature map and thus needs to be adapted. However, this is not the case for the activation layers, nor is this necessary for pooling operations. Finally, our framework is also capable of tracking the feature map channels when concatenating or stacking multiple feature maps together, and thus allows to prune these convolutions as well. One limitation of our framework is that we cannot prune convolutions whose feature maps are used in element-wise operations with other feature maps, such as residual connections. Indeed, pruning these convolutions would require to prune the same channels from both convolutions and is not currently implemented in our framework.

Once the dependency tree has been built, the actually pruning starts. Our iterative pruning pipeline is described in pseudo code in Algorithm 1. We iteratively prune X% of our model, and then retrain it for a maximum number of epochs *E*, in order to reach the same accuracy as the original model. Note that the absolute number of pruned channels at each step diminishes over time, because the network itself becomes smaller. In order to limit the total runtime of the algorithm, we set a hard limit to the minimal number of channels to prune. Once the algorithm prunes less than 5 channels of the network per step, the pruning stops. If the new model reaches an accuracy during retraining that is α higher than the original accuracy, training is stopped prematurely. This prevents overfitting on the validation set and allows to set a higher value for the number of epochs *E*, without having unnecessary long retraining times for the first few iterations of the pruning algorithm. Finally, if we cannot reach $AP_{original}+\alpha$ after retraining for *E* epochs, we still continue our pruning pipeline if the accuracy is only β below the original accuracy. The rationale behind this is that usually our validation set is small, and thus a minor drop in accuracy on this dataset might not be representative for the entire data. We therefore set the lower bound slightly below the original validation accuracy. Note that we use a separate validation set with this pruning algorithm, to prevent overfitting on our test set. Only the final models after pruning are tested on the test set and compared with the original models for verification.
**Algorithm 1**: Pruning pipeline.
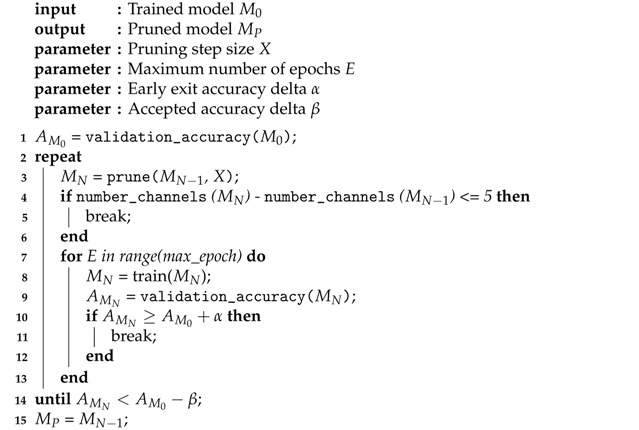


Since pruning is a very active research topic, there are a wide range of techniques which select the appropriate channels to be removed [31]. In this paper, we aim to study how relatively simple approaches translate to operational use-cases. We thus only implemented and compared two different pruning techniques, based on the L2-norm [13] and the Geometric Median (GM) [14]. The L2-norm based pruning technique is straightforward: we compute the L2-norm of the weights of each prunable convolutional channel in the network. The channels with the lowest L2-norm are considered to be the least important and are thus pruned. However, the scale of the L2-norm can vary significantly depending on the depth of the convolution in the network. Inspired by Molchanov et al. [13], we further normalize the L2-norm of each channel in a convolution:(4)$$Importance\left(w_{i}\right)=\frac{\left|\right|w_{i}{\left|\right|}_{2}}{\sqrt{\sum_{w_{j}\in W}\left|\right|w_{j}{\left|\right|}_{2}^{2}}}$$
where $w_{i}$ and $w_{j}$ are the weights of a single channel in a convolution *W*. This allows us to compare the importance of channels in different convolutions and thus allows us to prune a certain percentage of the channels of all convolutions in the network.

In 2019, He et al. [14] discussed two issues related to norm-based pruning:Small Norm Deviation: The different norms might be concentrated in a small interval, which makes it hard to select an optimal threshold for pruning channels.Large Minimum Norm: The channels with minimum norm may not be arbitrarily small. In this case, channels which we consider least important might still contain relevant information and pruning them might have negative consequences.

In order to solve these problems, they propose a method to prune channels that are closest to the geometric median of all channels in a convolution. After optimization, the final formula for the importance of a channel in a convolution is given as:(5)$$Importance\left(w_{i}\right)=\sum_{w_{j}\in W}\left|\right|w_{i}-w_{j}{\left|\right|}_{2}$$

A potential disadvantage of this technique is that it becomes impossible to compare channels of different convolutions, as they have vastly different importance values. As such, the geometric median can only be used on a per-layer basis and thus we can only prune all the layers in our network uniformly. In order to mitigate this, in this paper we propose to combine both L2-based pruning and GM-based pruning (e.g., pruning 5% with L2 and 5% with GM).

### 2.4. Post-Training Quantization

The final step in our optimization pipeline is post-training quantization. Changing the weights of the model from 32-bit floating point (FP32) to 16-bit floating point (FP16) indeed drastically reduces the memory usage and computation time on embedded devices. Methods exist to transform weights even further to 8-bit integers. However, as we speak they are not yet available for our targeted platform (CUDA) in PyTorch v.1.7.0.

We ran our final benchmark on the Nvidia Jetson AGX Xavier, by re-implementing our entire inference pipeline in C++ with LibTorch, the underlying library for PyTorch [29]. Since this entire pipeline is implemented with LibTorch tensors, it allows us to run the pre-processing, model and post-processing on either CPU or GPU in FP32 or FP16.

## 3. Results

In this section we discuss the results of our experiments. We optimized the YoloV2 network as detailed in Section 2 for an academic dataset (see Section 3.1) and an operational dataset (see Section 3.2).

### 3.1. Academic Dataset: Pascal VOC

This section covers the experiments conducted on the Pascal VOC dataset. As the original YoloV2 network was specifically designed for this dataset, we do not expect a reduction in computational cost by a significant margin, without affecting the accuracy. However, it is worth trying to optimize it further.

#### 3.1.1. Training on Pascal VOC

We train our models on the training dataset of Pascal VOC, using the same hyperparameters discussed by Redmon and Farhadi [4] for all of our models. As discussed in the original paper, we transfer learn our models from ImageNet [6] and train them for a total of 80,200 batches of 64 images each, again referring to the original implementation [4]. However, since we will be needing the validation dataset whilst pruning, we can only use the training data to actually train our model, as opposed to the original implementation, which combined both training and validation data together for training purposes. This results in slightly lower accuracies overall, but is paramount for ensuring we do not overfit our model on the test data whilst pruning.

Figure 7 shows the accuracy in mean Average Precision (mAP) and the model complexity in number of Multiply Accumulate operations (MAC). Our MobileYoloV2 network, which uses depth-wise separable convolutions, requires almost four times fewer computations compared to the original YoloV2. However, it only reaches an mAP of 55.9% on the validation set, which is almost 7% lower. Using our Upsample methodology, we manage to increase the validation mAP to 58.2%, at an extra computational cost of 6.1 GMAC. MobileYoloV2 Upsample thus requires around 1.5 times less computations than the original YoloV2 network. Using the same Upsample methodology on the regular YoloV2 network also increases the accuracy and results of our best model, reaching 65.4% AP.

#### 3.1.2. Pruning on Pascal VOC

We iteratively prune X% of the channels of our model, as explained in Algorithm 1, with parameters α=3 and β=2. During the training stage, we train for a maximum number of 10,000 batches with a fixed learning rate of 0.0001. For each of the methods, we use three different step sizes: 5%, 10% and 15%. Finally, we also combine both methods, pruning 5% with the combination of both L2 and GM, pruning 5% with L2 and 10% with GM and pruning 10% with L2 and 5% with GM. The results of these experiments are shown in Figure 8. We select the best method for each network by looking at the number of MAC-operations in the network. These are shown in Table 4.

Even though YoloV2 was designed for this academic dataset, we still manage to prune over 50% of the computations, without dropping accuracy on the validation data. By pruning MobileYoloV2, we manage to reduce the computations with a factor of 2.5, requiring only 1.5 GMAC. Compared to the original YoloV2 network, this is a reduction in computational complexity of a factor of 9.8. Pruning on this dataset works best with smaller step sizes, which results in long pruning times, since we remove fewer channels on each iteration.

An important thing to note is that all networks are pruned more with the L2 method. Figure 9 shows the number of pruned channels per layer for YoloV2. With the L2 method, we pruned up to 60% of the middle and late layers, whilst only pruning 10–20% of the early layers. This results in a higher total pruning rate than the GM method, which is only able to prune all layers uniformly.

#### 3.1.3. Quantization on Pascal VOC

Once we selected our different pruned networks, we can perform a final benchmark on our test set. As explained in Section 2.4, we run our benchmark on an NVidia Jetson AGX Xavier, using Libtorch, the C++ interface of PyTorch. When computing the average precision, we set the confidence threshold of our models to 1%, in order to capture the full range of detections our models return. However, when performing timing benchmarks, selecting a proper confidence threshold has an influence on the post-processing time. We therefore select the confidence of the best $F_{1}$ point of our Precision-Recall (PR) curve. As Pascal VOC is a multi-class dataset, we need to compute a micro-averaged PR curve in order to compute a single $F_{1}$ point and confidence value, which are shown in Figure 10.

The results of this benchmark are given in Table 5. YoloV2 Upsample reaches the highest accuracy, with 65.2% AP, but is also the slowest network with an inference time of 52 ms per image. MobileYoloV2 is the fastest model with an inference time of 15 ms, but only reaches an accuracy of 56%. MobileYoloV2 Upsample is 8 ms slower, with an inference time of 23 ms. This is still almost two times faster than regular YoloV2, but it does not reach the same accuracy and is still 5% below the original, with an accuracy of 58%. We can also see that the post-processing is quite stable and takes around 6 ms for all the networks.

Pruning the models results in a major speedup of almost a factor of 2, except for the MobileYoloV2 network, which barely got any speedup at all. However, it still remains the fastest FP32 network with an inference latency of 13 ms.

Finally, we also ran our models in FP16 mode. The first thing to note is that switching to FP16 does not seem to influence the accuracy at all, whilst decreasing the latency by a significant margin for most networks. In this benchmark, it seems that the faster mobile networks do not have as big of a speedup as compared to the two other models. Still, our fastest network overall is Regular MobileYoloV2 Upsample in FP16 mode, which has an inference time of 17 ms (with post-processing) and can thus process an image stream of up to 60 Frames per Second (FPS).

Table 5 shows that pruning results in only a minor decrease of AP compared to the regular networks. The micro-averaged PR curve of Figure 10 further shows that this decrease in AP comes from a faster decline in recall more specifically.

### 3.2. Operational Dataset: LWIR Railway Surveillance Data

#### 3.2.1. Training on LWIR

Gaus et al. [9] proved that it is beneficial to use a pretrained model on RGB data, and subsequently apply transfer learning in order to fine-tune the model on a smaller IR dataset. We thus follow the same paradigm and start from the same pretrained ImageNet weights as before. We manually fine-tune some of the hyperparameters in order to reach a better accuracy with YoloV2 on the validation data. We then proceed to train the other networks with these fine-tuned hyperparameters (found in Table 6).

Figure 11 shows the PR curves and AP accuracy of the trained models, as well as their computational complexity in number of MAC-operations. Similarly to the VOC training, MobileYoloV2 requires almost four times less computations. However, as this is a less complex dataset, MobileYoloV2 reaches an accuracy of 81.35% AP, which is only −1.35% lower than the original. Using the Upsample approach, we can again increase the accuracy. Hence MobileYoloV2 Upsample reaches an accuracy of 84.95% AP, beating the original YoloV2 by a margin of +2.25%, whilst only requiring $\frac{2}{3}$ of the computations.

#### 3.2.2. Pruning on LWIR

We iteratively prune in the same manner as explained above, but only allow a maximum number of 5000 batches of retraining. Because this dataset is less challenging, we also increase the different step sizes to 10%, 15% and 20% for both methods individually. When combining the methods together, we try pruning with step sizes of 5 + 5%, 5 + 10%, 10 + 5% and 10 + 10%. The results of these experiments are shown in Figure 12 and the best pruned versions of each network are found in Table 7.

Amazingly, we see that on this LWIR dataset, we manage to prune more than 95% of the computations. Our smallest model, MobileYoloV2 Upsample requires 233 times fewer computations after pruning and 349 times fewer computations than the original YoloV2, whilst achieving the same AP on the validation dataset. This clearly proves our hypothesis that these neural networks are vastly oversized for simpler, constrained computer vision problems as often found in real-life and industrial applications.

A current limitation of our pruning pipeline implementation is that we cannot track the feature map dependencies after the *“reorg”* layer. This can be seen in Figure 13, where layers 25 and 26 are not pruned at all. By swapping this layer for an *“upsample”* layer, we effectively allow to prune these two layers as well, resulting in an even higher pruning rate on this dataset.

Looking at Figure 12, we note that combining both the L2 and GM pruning methods yields the best results overall, followed by GM. L2 gives the worst pruning results. Figure 13 shows that we prune all layers more evenly on this dataset and thus the downside of using GM, which can only prune all layers uniformly, is not an issue here. In fact, the GM pruning method seems to be a more powerful pruning technique than L2 for this dataset.

#### 3.2.3. Quantization on LWIR

Finally, we select the best pruned networks for each model and run a benchmark on the NVidia Jetson AGX Xavier in C++. The results of this benchmark can be found in Table 8. Just as for the Pascal VOC benchmark, we set the confidence threshold to 1% when measuring the AP and we select the best F1 working point for measuring inference runtime.

YoloV2 Upsample is the best, yet slowest model, reaching an AP of 68.2%, for a latency of 80 ms per image on this dataset. MobileYoloV2 is the fastest regular network at 24 ms latency, but drops 6.7% AP compared to the original YoloV2. In comparison, MobileYoloV2 Upsample, with a latency of 37 ms, is still 1.8 times faster than the original YoloV2 whilst reaching a similar accuracy as well.

When looking at the pruned networks, we can see that the fastest network is YoloV2 Upsample, with a latency of 4.4 ms, even though Table 7 shows it has more computations than MobileYoloV2 Upsample. Splitting a convolution in a depth-wise and point-wise part results in two times more layers in the network, but each layer is computationally less expensive. However, as we pruned up to 99% of all channels in the network, the computational advantage becomes much smaller and the overhead of having more layers becomes much more noticeable, which is why YoloV2 and YoloV2 Upsample are faster on this dataset when pruned.

Finally, switching to FP16 for our computations results in a speedup of all networks, without affecting the accuracy. Pruned YoloV2 seems to be the fastest network in this setting, but not by much compared to pruned YoloV2 Upsample. Furthermore, YoloV2 Upsample has a 5% better accuracy and would thus most likely be the preferred network. With a combined latency of 7.8 ms, this network can handle image streams of up to 128 FPS.

Looking at the PR curves in Figure 14, we can again conclude that the major difference between the regular and pruned models is that the recall of the pruned models declines faster. Even MobileYoloV2 Upsample, which starts at a higher recall level, finally has a big drop in recall and shoots below the original model.

## 4. Discussion

Table 9 shows a summary of our experimental results in the form of an ablation study. Our experiments showed that swapping out regular convolutions for depth-wise and point-wise convolutions is a good technique for reducing the computational complexity and thus inference latency of a model. Contrary to the results shown by Howard et al. [11], this modification unfortunately harms the accuracy of our models by a significant margin. Indeed, Table 9 shows that this optimization requires 3.9 times less computations than YoloV2, but also shows a decrease of 7.4% AP on the Pascal VOC test dataset and 6.2% on the LWIR test dataset. Using the Upsample trick that we propose in this paper, we manage to reduce this difference in AP to only 4.9% for Pascal VOC. On the LWIR dataset, we manage to completely eliminate this difference and beat the original model by a margin of 0.4% on the test data (*“+DW+UP”* in Table 9). These results also demonstrate that the constrained setting of our operational dataset allows us to use lighter models with less modeling capabilities. While we did manage to use this technique successfully, one might argue that this technique is not automatic and is in fact a model engineering trick. Still, as we did not redesign a network from scratch, but simply swapped out all convolutions, we believe this technique to be justified in this article.

Pruning models allows for a completely automatic modification of a network and gives good results on both datasets, as we are able to prune away more than 50% of the computations of our models for Pascal VOC and over 95% for the LWIR dataset. The lesser results on Pascal VOC were to be expected, as the original networks were specifically designed to work on such a challenging dataset. We still managed to prune a significant amount of computations for this dataset, resulting in an inference speedup factor of two for the YoloV2 model. These results might point towards the fact that the models need complex modeling capabilities for training, but can be reduced significantly afterwards. Combined with depth-wise separable convolutions, our fastest network is MobileYoloV2, which contains 9.8 times fewer computations and is 3.3 times faster than the original YoloV2. For the case of the LWIR dataset, we managed to prune over 95% of the networks, demonstrating that these network designs are indeed usually oversized because they are not targeted at these more constrained operational scenarios. Our fastest pruned network, Yolov2 Upsample, is 85 times less computationally expensive and more than 15 times faster than the original YoloV2. Surprisingly, this was not the network with the least amount of computations. Indeed, MobileYoloV2 Upsample requires even less computations and is 349 times less computationally expensive than YoloV2. However, the overhead of having more layers makes this network slower on a GPU, but we expect this network to have a bigger speedup factor when using only CPU or when implemented on custom hardware. Even though geometric median pruning yields better results than L2-based pruning on our operational dataset, the results on Pascal VOC seem to indicate that it might be beneficial to introduce some form of normalization, which would allow to prune across layers with this technique.

Finally, we observe that post-training quantization really is a must when deploying models on devices with limited computational resources. Overall, it results in an average speedup factor of two, without any noticeable difference in accuracy. The speedup factor does seem to be less for our faster mobile networks, as they contain less computations overall.

While simply combining all optimization techniques together does indeed yield faster results than the individual techniques, our ablation study in Table 9 demonstrates that this is not always the optimal solution. When deploying this optimization pipeline on an operational use-case, it might thus be beneficial to carefully try the different techniques, when you need the best possible trade-off. Doing so on our operational dataset, we created a pipeline that is able to handle a 640x512 image stream at a framerate of up to 128 FPS on an NVidia Jetson AGX Xavier. Moreover, it increased the accuracy by a margin of 5% AP compared to the original YoloV2 network, which can only process images up to 14 FPS. All the code used for training, pruning and finally benchmarking our models has been released in our open source library, Lightnet [25].

For the hypothesis we presented earlier in this paper, “The more constrained a problem is, the more the network can be optimized”, our experimental results in Table 9 show clear evidence. The achieved factors, both in compute reduction and speed-up, are orders of magnitude larger on our constrained LWIR problem as compared to the challenging Pascal VOC problem. Other recent publications substantiate this claim, although they do not explicitly compare problems with different constrainedness. The single-shot object detector pruning experiments reported in Wu et al. [18] indeed resulted in a large optimization factor as well (32× reduction in number of parameters, 20× smaller model size). This coincides with their problem, which is similarly greatly constrained: apple flowers have a very limited amount of intra-class variance, and the background is very similar in all dataset images. Another example is the work of Ayob et al. [19], combining depth-wise separable convolutions and pruning for underwater object detection from a static aquarium camera, which resulted in a reduction with a factor of 161 in model size and a speed-up with a factor of 4.7. Although the exact numbers are not comparable with our results, where we even combine more optimization techniques, these studies show indeed that for constrained problems great optimization factors can be achieved. Actually, our results put the findings of such studies in perspective, as from our experiments we know that the nature of the dataset itself greatly influences the optimization factors that can be achieved. It would have been more honest if such papers, presenting results on network optimization, would also report their achieved optimization factor on a standard academic dataset, and not only on their own operational use case.

## 5. Conclusions

In this paper, we investigated the achievable amount of optimization for object detection neural networks trained on different datasets. We observe that many real-life and industrial application scenarios of object detection are somehow constrained, in the sense that they e.g. have less object and background variation, less object classes to be detected or are limited to restricted or even fixed viewpoints and color schemes. This is in sharp contrast to the academic datasets designed for object detection challenges, such as Pascal VOC and MS COCO, which are very challenging and unconstrained. The hypothesis in this paper is that the resulting best scoring neural network architectures for these academic datasets, for instance Faster-RCNN, Yolo and SSD, are actually largely oversized for industrial applications and hence have plenty of room for optimization. However, gathering enough data in order to train completely new models for operationally use-cases is often unfeasible and thus transfer learning provides a great mechanism to reduce the required amount of data. This does mean that the same models are used and thus there is a need to be able to adapt networks to different scenarios. We therefore build a pipeline in order to combine multiple optimization techniques together and test it, as a case study, on two distinct datasets (on opposites of the dataset complexity spectrum). In terms of computational complexity, we observed that we can safely reduce the model trained on the academic dataset by a factor of 9.8, while the model trained on the constrained operational dataset can be reduced by a factor of 348.8. In terms of model inference speed on a typical embedded GPU, we reached speed-up factors of 3.9 and 15.3, respectively. This indeed demonstrates that the potential of network optimization is much larger for a constrained object detection problem. Finally, our ablation study shows that simply combining all optimization techniques together does not necessarily yield the best results. While the full combination provides a better speed-accuracy trade-off than the individual optimization methods for both our test cases, we managed to get even better results by selecting only a subset of the techniques.

In the future, we want to look at more advanced pruning techniques and validate them on various different operational use-cases. Additionally, the runtime speed of our models could be reduced even further by using tools such as TensorRT, which generate highly optimized inference code by analyzing a model. It might thus be interesting to run our optimized and pruned models with TensorRT, as the reduced memory footprint of our models might allow the tool to select faster kernels and thus yield even more impressive results.

## Figures and Tables

**Figure 1 jimaging-07-00064-f001:**
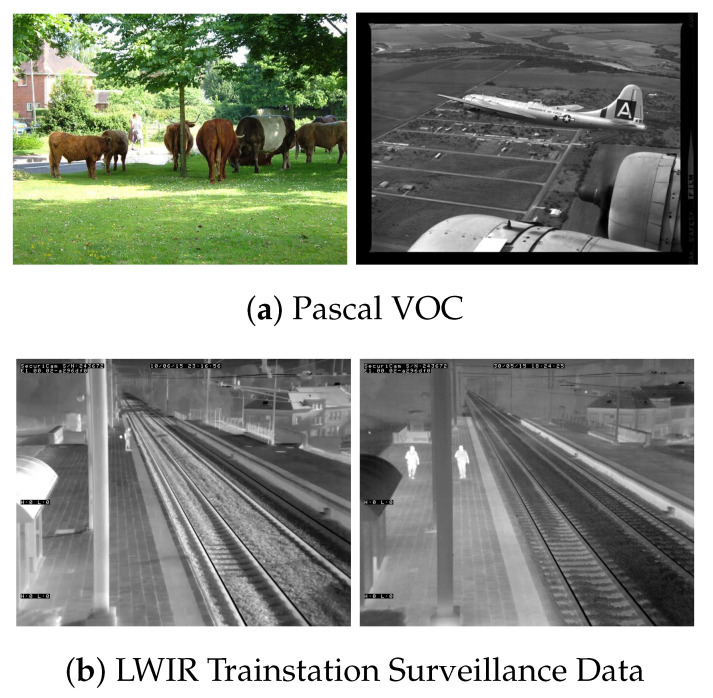
Example images from both datasets.

**Figure 2 jimaging-07-00064-f002:**
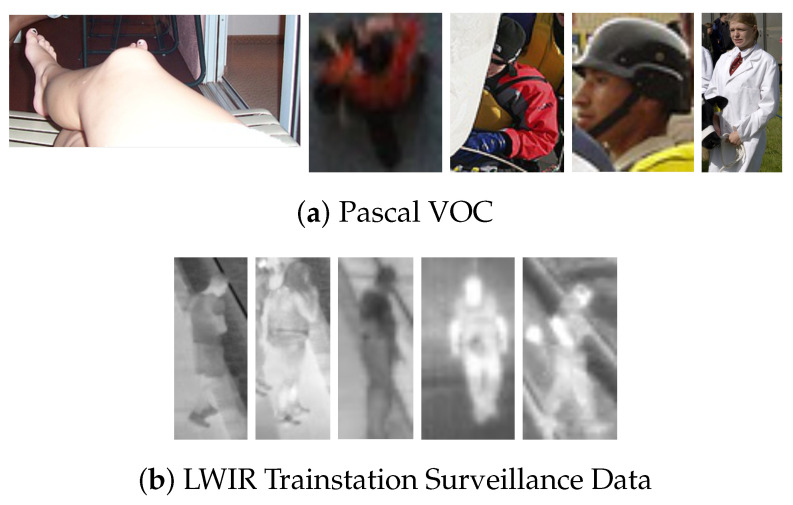
Bounding box cutouts of the “person” class for both datasets. Pascal VOC contains a lot of intra-class variance, whilst persons in the Long Wave Infrared (LWIR) dataset have a similar look.

**Figure 3 jimaging-07-00064-f003:**
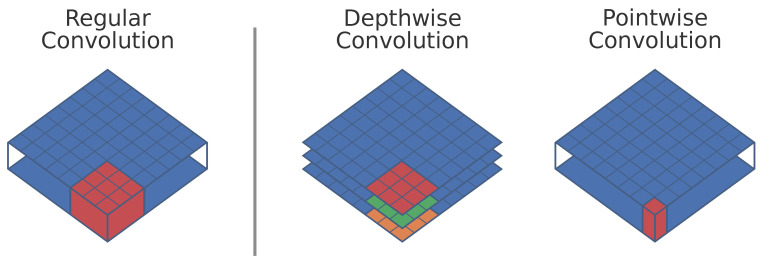
A regular convolution compared to a depth-wise and point-wise convolution. A regular convolution can be seen as a volumetric operation, whilst the depth-wise separable convolution can be considered as a sum of two 2D operations.

**Figure 4 jimaging-07-00064-f004:**
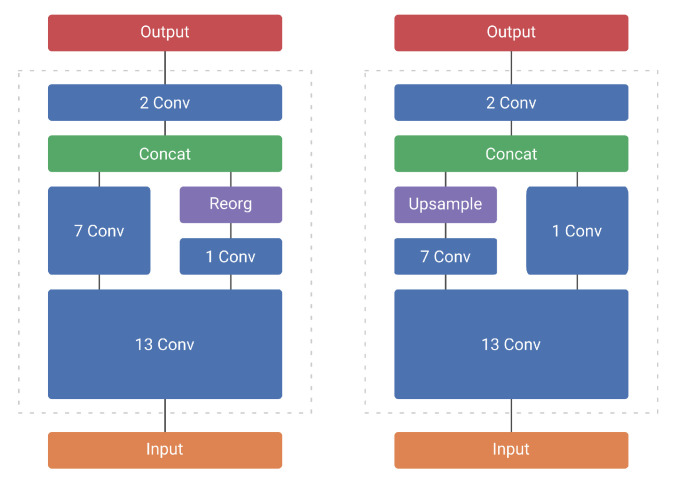
YoloV2 (**left**) and YoloV2 Upsample (**right**).

**Figure 5 jimaging-07-00064-f005:**
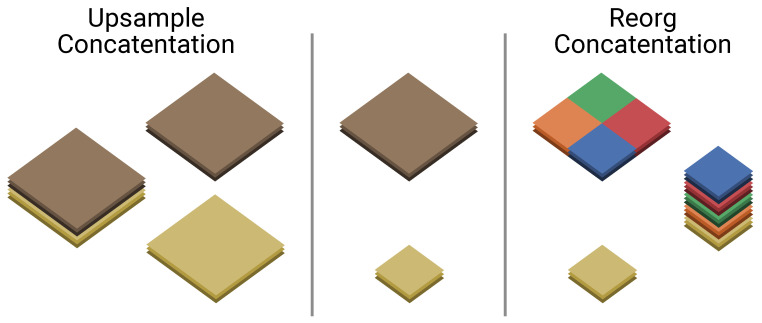
YoloV2 reorg technique vs our upsample technique for concatenation. The reorg technique splits the bigger feature map, whilst the upsample technique increases the resolution of the smaller feature map.

**Figure 6 jimaging-07-00064-f006:**
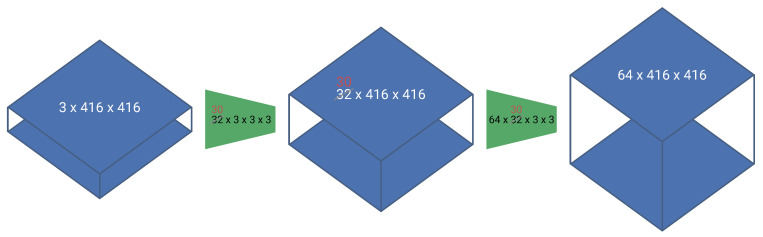
Dependency tracking when pruning convolutions. When reducing the number of channels of the first convolutional kernel (**green**), the number of output features (**blue**) changes as well. This means that we need to adapt any module that uses that feature map (i.c. the next convolution).

**Figure 7 jimaging-07-00064-f007:**
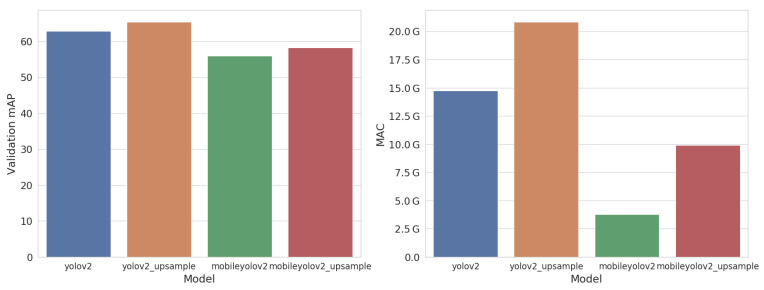
Accuracy (mean Average Precision (mAP)) and complexity (Multiply-Accumulate (MAC)) of our models on the Pascal VOC validation dataset.

**Figure 8 jimaging-07-00064-f008:**
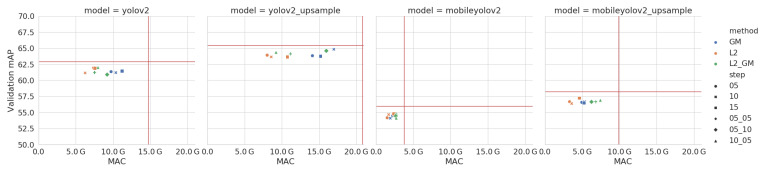
Accuracy (mAP) vs computational complexity (MAC) of the pruned models on the Pascal VOC validation data. Note the red horizontal and vertical lines, which indicate the values of the unpruned models. The Y-axis does not start at zero, in order to more clearly show the small differences in accuracy.

**Figure 9 jimaging-07-00064-f009:**
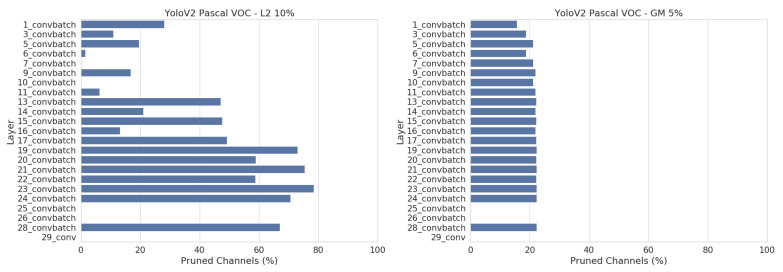
Number of discarded channels per layer after pruning YoloV2 with L2 (**left**) and GM (**right**). The best pruned version of each are shown. These plots show the disadvantage of GM pruning, which can only prune all layers uniformly and thus gets a lower overall pruning percentage.

**Figure 10 jimaging-07-00064-f010:**
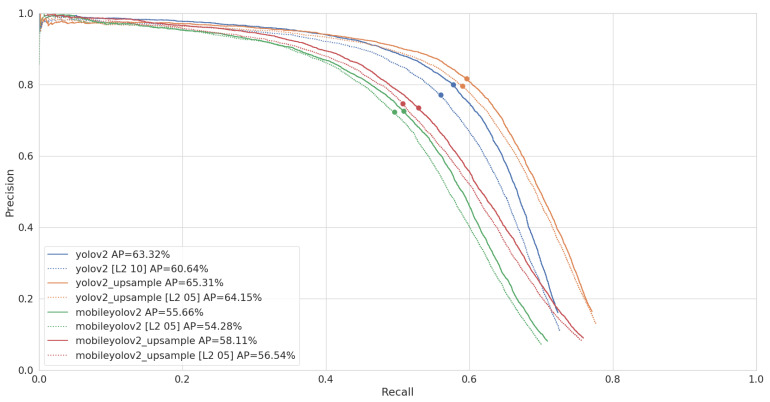
Micro-averaged Precision-Recall (PR) curve and AP of our models on the Pascal VOC test set. The best F1 point is marked for each curve, which is used as a confidence threshold when performing the timing benchmarks.

**Figure 11 jimaging-07-00064-f011:**
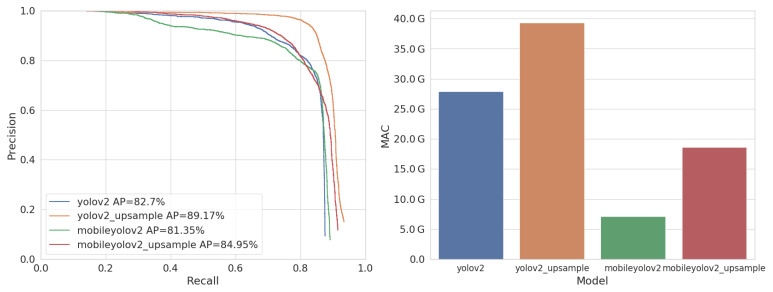
Accuracy (AP) and complexity (MAC) of our models on the LWIR Railway Surveillance validation dataset.

**Figure 12 jimaging-07-00064-f012:**
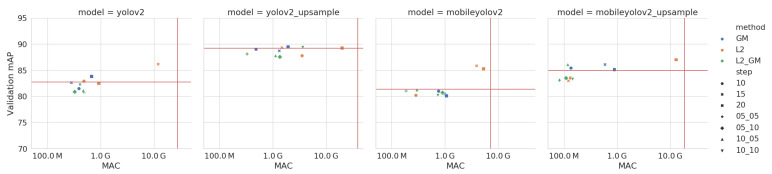
Accuracy (AP) vs computational complexity (MAC) of the pruned models on the LWIR Railway Surveillance validation data. Note the red horizontal and vertical lines, which indicate the values of the unpruned models. The Y-axis does not start at zero, in order to more clearly show the small differences in accuracy and the X-axis is on a logarithmic scale.

**Figure 13 jimaging-07-00064-f013:**
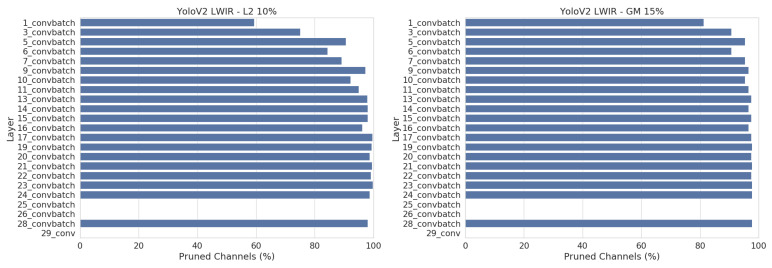
Number of pruned channels per layer after pruning YoloV2 on LWIR with L2 (**left**) and GM (**right**). The best pruned version of each are shown. Compared to Figure 9, the disadvantage of uniform pruning is less of an issue on this dataset, and thus GM pruning manages to outperform L2 pruning.

**Figure 14 jimaging-07-00064-f014:**
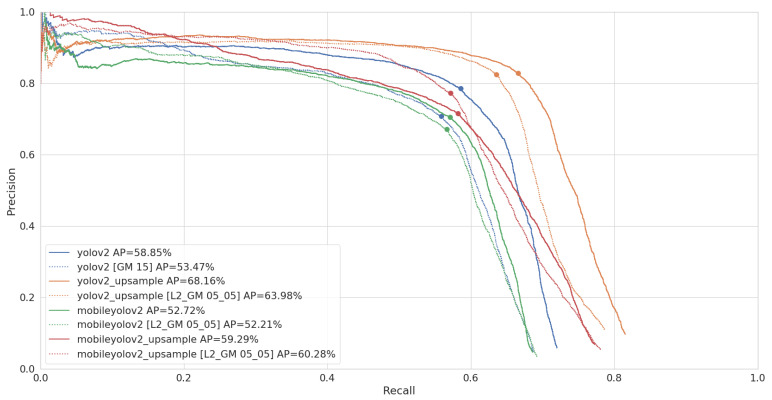
PR curve and AP of our models on the LWIR test set. The best F1 point is marked for each curve, which is used as a confidence threshold when performing the timing benchmarks.

**Table 1 jimaging-07-00064-t001:** Number of images per dataset.

Split	VOC	LWIR
Train	14,041	13,649
Validation	2510	3142
Test	4952	5061
Total	21,503	21,852

**Table 2 jimaging-07-00064-t002:** LWIR video sequence in each split. Note that there is no sequence 2 available in the dataset.

Train	0, 4, 7, 10, 12, 14, 16, 17, 18, 19, 20, 21, 22, 23, 24, 25, 26, 27
Validation	1, 5, 8
Test	3, 6, 9, 11, 13, 15

**Table 3 jimaging-07-00064-t003:** Complexity of YoloV2 and our proposed variants on Pascal VOC.

Model	Input Resolution	Output Resolution	FLOPs (GMAC)
YoloV2	416 × 416	13 × 13	14.74
YoloV2	832 × 832	26 × 26	58.96
YoloV2 Upsample	416 × 416	26 × 26	20.85
MobileYoloV2	416 × 416	13 × 13	3.80
MobileYoloV2	832 × 832	26 × 26	15.18
MobileYoloV2 Upsample	416 × 416	26 × 26	9.91

**Table 4 jimaging-07-00064-t004:** Accuracy (mAP) and complexity (GMAC) of the regular models, compared with the best pruned version of each model. Accuracy is computed on the Pascal VOC validation dataset.

Model	Regular		Pruned				
	mAP_val_ (%)	GMAC	Method	Step (%)	mAP_val_ (%)	GMAC	Reduction
YoloV2	62.9	14.7	L2	10	61.1	6.3	÷ 2.3
YoloV2 Upsample	65.4	20.9	L2	5	63.9	8.0	÷ 2.6
MobileYoloV2	55.9	3.8	L2	5	54.2	1.5	÷ 2.5
MobileYoloV2 Upsample	58.2	9.9	L2	5	56.7	3.3	÷ 3

**Table 5 jimaging-07-00064-t005:** Model accuracy and inference times measured on the NVidia Jetson Xavier and averaged over the Pascal VOC test set. Average Precision (AP) is measured with a threshold of 1% and timing results are measured with a threshold set at the best $F_{1}$ score. The best values are shown in bold.

	YoloV2		YoloV2 Upsample		MobileYoloV2		MobileYoloV2 Upsample	
	Regular	Pruned	Regular	Pruned	Regular	Pruned	Regular	Pruned
Best F_1_ (%)	67.1	64.9	**68.9**	67.8	59.8	58.8	61.5	60.4
Threshold (%)	45.8	45.3	47.7	46.9	39.8	39.8	36.1	39.5
GPU FP32								
AP_test_ (%)	63.0	60.6	**65.2**	64.2	55.6	54.0	58.1	56.3
Model (ms)	43.83	22.42	52.06	25.64	15.37	**13.22**	23.66	13.34
Post (ms)	6.77	6.40	6.71	6.45	5.94	6.13	6.49	5.80
GPU FP16								
AP_test_ (%)	63.0	60.6	**65.2**	64.2	55.7	54.0	58.1	56.3
Model (ms)	16.92	12.54	17.92	13.17	11.94	11.37	**11.24**	12.53
Post (ms)	6.19	5.86	6.25	5.80	5.97	6.16	5.50	6.03

**Table 6 jimaging-07-00064-t006:** Modified hyperparameters for LWIR training. Any parameter that is not listed here is the same as for the VOC training.

Hyperparameter	VOC	LWIR
Input Dimension	416 × 416	640 × 512
Max Batches	80,200	25,000
Object Scale	5.0	5.0
No-object Scale	1.0	1.0
Coord Scale	1.0	2.0
Learning Rate	0.001	0.001
	[×0.1 after batch 40,000 and 60,000]	[×0.1 after batch 10,000 and 17,000]

**Table 7 jimaging-07-00064-t007:** Accuracy (AP) and complexity (GMAC) of the regular models, compared with the best pruned version of each model. Accuracy is computed on the LWIR Railway Surveillance validation dataset.

Model	Regular		Pruned				
	AP_val_ (%)	GMAC	Method	Step (%)	AP_val_ (%)	GMAC	Reduction
YoloV2	82.7	27.9	GM	15	82.6	0.28	÷ 99.6
YoloV2 Upsample	89.2	39.4	L2_GM	5 5	88.1	0.33	÷ 119.4
MobileYoloV2	81.3	7.2	L2_GM	5 5	81.0	0.18	÷ 40.0
MobileYoloV2 Upsample	85.0	18.6	L2_GM	5 5	83.1	0.08	÷ 232.5

**Table 8 jimaging-07-00064-t008:** Model accuracy and inference times measured on the NVidia Jetson Xavier and averaged over the LWIR test set. AP is measured with a threshold of 1% and timing results are measured with a threshold set at the best F1 score. The best values are shown in bold.

	YoloV2		YoloV2 Upsample		MobileYoloV2		MobileYoloV2 Upsample	
	Regular	Pruned	Regular	Pruned	Regular	Pruned	Regular	Pruned
Best F_1_ (%)	67.1	62.5	**73.8**	71.8	63.1	61.5	64.2	65.7
Threshold (%)	39.6	37.5	49.6	40.7	36.5	33.8	41.2	43.1
GPU FP32								
AP_test_ (%)	58.9	53.5	**68.2**	64.0	52.7	52.2	59.3	60.3
Model (ms)	65.93	6.63	79.34	**4.37**	24.04	7.20	36.94	5.86
Post (ms)	3.51	3.63	3.71	3.61	3.68	3.55	3.53	3.52
GPU FP16								
AP_test_ (%)	58.9	53.5	**68.2**	63.8	52.8	51.8	59.5	60.2
Model (ms)	21.58	**4.25**	25.86	4.37	9.87	5.50	14.13	6.49
Post (ms)	3.48	3.45	3.66	3.43	3.71	3.50	3.57	3.37

**Table 9 jimaging-07-00064-t009:** Ablation study of the different optimization methods, evaluated on both the Pascal VOC and LWIR test dataset. Note that the inference time is not including the post-processing. The best values are shown in bold.

Model	Pascal VOC			LWIR		
	mAP (%)	GMAC	Inference (ms)	AP (%)	GMAC	Inference (ms)
YoloV2	63.0	14.7	43.8	58.9	27.9	65.9
+DW	55.6 (−7.4)	3.8 (÷3.9)	15.4 (÷2.8)	52.7 (−6.2)	7.2 (÷3.9)	24.0 (÷2.7)
+UP	**65.2** (+2.2)	20.9 (×1.4)	52.1 (×1.2)	**68.2** (+9.3)	39.4 (×1.4)	79.3 (×1.2)
+PRUNE	60.6 (−2.4)	6.3 (÷2.3)	22.4 (÷2.0)	53.5 (−5.4)	0.28 (÷99.6)	6.6 (÷10)
+QUANT	63.0 (−0.0)	14.7 (÷1.0)	16.9 (÷2.6)	58.9 (−0.0)	27.9 (÷1.0)	21.6 (÷3.1)
+DW +UP	58.1 (−4.9)	9.9 (÷1.5)	23.7 (÷1.8)	59.3 (+0.4)	18.6 (÷1.5)	36.9 (÷1.8)
+DW +PRUNE	54.0 (−9.0)	**1.5** (÷9.8)	13.2 (÷3.3)	52.2 (−6.7)	0.18 (÷155)	7.2 (÷9.2)
+DW +QUANT	55.7 (−7.3)	3.8 (÷3.9)	11.9 (÷3.7)	52.8 (−6.1)	7.2 (÷3.9)	9.9 (÷6.7)
+UP +PRUNE	64.2 (+1.2)	8.0 (÷1.8)	25.6 (÷1.7)	64.0 (+5.1)	0.33 (÷84.5)	4.4 (÷15.0)
+UP +QUANT	**65.2** (+2.2)	20.9 (×1.4)	17.9 (÷2.4)	**68.2** (+9.3)	39.4 (×1.4)	25.9 (÷2.5)
+PRUNE +QUANT	60.6 (−2.4)	6.3 (÷2.3)	12.5 (÷3.5)	53.5 (−5.4)	0.28 (÷99.6)	**4.3** (÷15.3)
+DW +UP +PRUNE	56.3 (−6.7)	3.3 (÷4.5)	13.3 (÷3.3)	60.3 (+1.4)	**0.08** (÷348.8)	5.9 (÷11.2)
+DW +UP +QUANT	58.1 (−4.9)	9.9 (÷1.5)	**11.2** (÷3.9)	59.5 (+0.6)	18.6 (÷1.5)	14.1 (÷4.7)
+DW +PRUNE +QUANT	54.0 (−9.0)	**1.5** (÷9.8)	11.4 (÷3.8)	51.8 (−7.1)	0.18 (÷155)	5.5 (÷12.0)
+UP +PRUNE +QUANT	64.2 (+1.2)	8.0 (÷1.8)	13.2 (÷3.3)	63.8 (+4.9)	0.33 (÷84.5)	4.4 (÷15.0)
+DW +UP +PRUNE +QUANT	56.3 (−6.7)	3.3 (÷4.5)	12.5 (÷3.5)	60.2 (+1.3)	**0.08** (÷348.8)	6.5 (÷10.1)

## Data Availability

Publicly available datasets were analyzed in this study. The Pascal VOC data can be found here: http://host.robots.ox.ac.uk/pascal/VOC. The LWIR Railway Surveillance data can be found here: https://iiw.kuleuven.be/onderzoek/eavise/viper/dataset. The code used in this study can be found at https://gitlab.com/eavise/lightnet and https://gitlab.com/EAVISE/top/voc (accessed on 31 March 2021).

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
