# Peer review of "Investigating the Potential of Network Optimization for a Constrained Object Detection Problem"

_2313-433X, 2021, doi:10.3390/jimaging7040064_

Round 1

Reviewer 1 Report

The article "Optimizing Existing Networks for Fast Inference in Constrained Object Detection Problems" deals with the tuning of deep neural networks for object recognition tasks. The central topic is the reduction of the network size in existing neural networks trained on more difficult data sets. The contribution is experimentally supported by an extremely high speed-up factor.

In my opinion, the research methodology is unsound and in important aspects not informed by the literature. The article is full of bizarre statements and arguments. For example, the article starts with criticising that current methods are too "complicated" and evaluated on too "challenging academic datasets". But to the reader, this could just mean that the paper deals with an oldfashioned, already solved problem. The authors remark that the methods used to solve those challenging problems are too slow to solve the easier problems. While this is probably true, it could also indicate that the authors simply try to use the wrong methods to solve a simple problem. This impression is confirmed in a discussion of constraints on scene properties, where the authors consider to use methods for relatively unconstrained scenes to solve highly constrained scenes. This is not a convincing line of arguments and motivation of the paper.

However, it is known that some pre-trained networks can be used successfully in related applications, a methodology known as transfer learning. This aspect is, however, not discussed in the article. To look at the problem from this angle would require a consideration of why the problem at hand is related to the problem solved by any existing network. This has not been done systematically in the article, which again confirms the impression that the authors apply the wrong method too their problem.

The literature review about pruning algorithms and other simplifications is inadequate. First, it is too short (just three lines). Secondly, it is criticised that the existing methods are tested on the "same academic datasets" that they have been trained for. This, however, is an entirely consistent and sound procedure, so it should not be criticised. Instead, the authors need to state their motivation and procedure in a different way.

The authors go on to transfer a model trained on the Pascal VOC dataset to the LWIR data set. These datasets are quite different in complexity. Even the spectrum of light is different. There should be a discussion of why this is a sensible procedure. If the task was just to process the lower complexity dataset, it would be fair to consider competitive algorithms designed for that type of problem. 

The article presents three methods to simplify the network architecture. But separable convolutions are already known from the literature. The proposed pruning algorithm seems quite straightforward and there is no discussion of how it relates to state of the art pruning algorithms. Much of the description sounds like an adjustment of the network architecture to a specific application problem, which is a standard consideration in neural networks. The process seems to be heuristical. There is no appreciation of the mathematical background, e.g. the manifold of the parameter space.

The authors achieve a high speedup, which is encouraging. But considering the awkward problem statement and methodology, a clear conclusion cannot be drawn. The authors need to think carefully about formulating a more convincing research hypothesis and connecting the research more sensibly to the literature.

Reviewer 2 Report

Overall, the paper is interesting and scientific sounded. However some points below need to be taken care of

  1. The writing can be improved, e.g. "like e.g. a fix....", basically "like" and "e.g." mean the same, so you don't need to have them together. Also, the word "like" is very informal for an academic paper. It can be replaced with some other words such as "such as", "for example" or "for instance". Furthermore, there are a lot of run-on sentences which makes it rather difficult to read. Tenses and choice of words should be revised. It can also be seen that the use of commas seems to be neglected sometimes . I recommend re-read the whole paper and amend it accordingly.
  2. The clearer explanation on why the selected datasets (over the other datasets available) allow you to evaluate and compare the efficiency of your proposed techniques should be given, it is rather vague in my opinion. Noted that I've read your comprehensive explanation of the datasets themselves and I am happy with that.
  3. Rather than just referring to the numbers/results listed in the tables by saying "see Table X" all the time, it is better to explicitly state those important numbers and explain them in the form of paragraphs. Those numbers should also be explained better.
  4. A comparison to other techniques rather than only to the focused technique, YoloV2, could be beneficial. I would rather see that provided in this paper as currently it seems like the comparison is a bit limited.

Reviewer 3 Report

The paper aims to improve the existing neural networks, especially CNN type of architectures, by reducing the computational burden. In particular, it considers separable convolutions which can be decomposed as depthwise convolution and pointwise convolution. A pruned convolution is also introduced. Then the modified YoloV2 network was tested on two data sets with performance comparison with the other related methods. First of all, the novelty of this work is very limited without any theoretical guarantees. Computational complexity could be derived and compared. Second, the assumption about the separability of convolution has certain restrictions in application. Narrowing generic convolutions down to separable ones may lose certain capabilities of a specific neural network. Moreover, it is unclear about major contributions of this paper. Lastly, the title is overclaiming since there is no evidence theoretically or empirically that this type of modifications can improve or even optimize all other existing networks. Only two sets of data are used, which will make ad hoc some of the proposed steps, e.g., convolution pruning. 

Minor issues include confusing figures without any explanations in the captions; unprofessional wording/terms, e.g., "the number of computations"; and missing details or references, e.g., line 156. 

Reviewer 4 Report

In this paper, three different techniques are presented for optimizing
YoloV2 runtime performance on Jetson AGX Xavier. The authors use
depthwise separable convolutions, pruning and weight quantisation in
order to accelerate inference and improve accuracy. Although this is
an interesting topic, I am not convinced about the novelty of this
paper:

(a) Nvidia provides TensorRT SDK
(https://developer.nvidia.com/tensorrt,
https://on-demand.gputechconf.com/gtc-eu/2017/presentation/23425-han-vanholder-efficient-inference-with-tensorrt.pdf)
for high-performance deep learning inference especially on Jetson
devices. The three methods described in this paper along with many
more (layer and tensor fusion, kernel tuning, dynamic tensor memory
etc) are already implemented. Moreover, there are some tools that can
perform this type of optimization almost automatically such as
torch2trt (https://github.com/NVIDIA-AI-IOT/torch2trt).

(b) It is a good practice to include all your contributions in the
introduction section with bullets. It will be easier for the reader to
understand your work as well as your contribution.

(c) A related work section is missing since deep learning optimization
seems to be a very active topic nowadays and no similar techniques are
presented.

(d) Also, no comparisons have been made with similar techniques. For
example, how does an optimized with TRT and your LightNet library
model perform on the same hardware?

(f) At line 408 there is a typo:  640x5126 =>  640x512 (?)

Round 2

Reviewer 1 Report

In the revision, major parts of the article have been rewritten. Most of the controversial statements have been removed, so the narrative is much more sound overall. The proposed method is of incremental originality at best. Most of the article, even in the methodology section, has aspects of a literature review. The proposed method, with all its heuristics and minor variations of known methods, is justified in detail. The discussion confirms and reflects upon the state of the art. I don't think the article will improve the reputation of the journal by much.

Reviewer 3 Report

The authors have made satisfactory revisions. However, the concerns about the novelty remain and some legends/plots could be made clearer or larger in figures 9-14.

Reviewer 4 Report

The latest version highlights major changes and improvements. The new
title does also seem more appropriate in the scope of the paper as it
converges more clearly in the aspect of network pruning and depth-wise
separable convolutions.

(a) However, as separable convolutions are well-known from the
literature, the authors should focus more on their network pruning
method. Comparisons with other state-of-the-art pruning algorithms are
also absent. For reference, the recent articles below implement
pruning techniques to various models including YOLO:

https://www.sciencedirect.com/science/article/pii/S0168169920318986
https://link.springer.com/chapter/10.1007/978-981-15-5281-6_7

(b) The authors aim at a generic pruning framework implementation, as
stated in the contributions paragraph (line 80). However, the
generalization of the presented technique is not sufficiently
supported in this paper. Consistent performance improvements by
experimenting with multiple object detection models, other than YOLO
itself, are required.

The conclussion is that the experimental results are insufficient and more comparisons are required.
